# Transitioning from Si to SiGe Nanowires as Thermoelectric Material in Silicon-Based Microgenerators

**DOI:** 10.3390/nano11020517

**Published:** 2021-02-18

**Authors:** Luis Fonseca, Inci Donmez-Noyan, Marc Dolcet, Denise Estrada-Wiese, Joaquin Santander, Marc Salleras, Gerard Gadea, Mercè Pacios, Jose-Manuel Sojo, Alex Morata, Albert Tarancon

**Affiliations:** 1Instituto de Microelectrónica de Barcelona, IMB-CNM (CSIC), C/Til·lers s/n—Campus UAB, Bellaterra, 08193 Barcelona, Spain; inci.donmez@gmail.com (I.D.-N.); marc.dolcet@e-campus.uab.cat (M.D.); denise.estrada@imb-cnm.csic.es (D.E.-W.); joaquin.santander@imb-cnm.csic.es (J.S.); marc.salleras@imb-cnm.csic.es (M.S.); 2Department of Advanced Materials for Energy Applications, Catalonia Institute for Energy Research (IREC), C/Jardí de les Dones de Negre 1, Planta 2, 08930 Barcelona, Spain; gerard.gadea.diez@gmail.com (G.G.); mercepacios@gmail.com (M.P.); jmsojo@irec.cat (J.-M.S.); amorata@irec.cat (A.M.); atarancon@irec.cat (A.T.); 3ICREA, Passeig Lluís Companys 23, 08010 Barcelona, Spain

**Keywords:** Silicon nanowires, SiGe nanowires, thermoelectricity, MEMS, energy harvesting, VLS-CVD

## Abstract

The thermoelectric performance of nanostructured low dimensional silicon and silicon-germanium has been functionally compared device-wise. The arrays of nanowires of both materials, grown by a VLS-CVD (Vapor-Liquid-Solid Chemical Vapor Deposition) method, have been monolithically integrated in a silicon micromachined structure in order to exploit the improved thermoelectric properties of nanostructured silicon-based materials. The device architecture helps to translate a vertically occurring temperature gradient into a lateral temperature difference across the nanowires. Such thermocouple is completed with a thin film metal leg in a unileg configuration. The device is operative on its own and can be largely replicated (and interconnected) using standard IC (Integrated Circuits) and MEMS (Micro-ElectroMechanical Systems) technologies. Despite SiGe nanowires devices show a lower Seebeck coefficient and a higher electrical resistance, they exhibit a much better performance leading to larger open circuit voltages and a larger overall power supply. This is possible due to the lower thermal conductance of the nanostructured SiGe ensemble that enables a much larger internal temperature difference for the same external thermal gradient. Indeed, power densities in the μW/cm^2^ could be obtained for such devices when resting on hot surfaces in the 50–200 °C range under natural convection even without the presence of a heat exchanger.

## 1. Introduction

The advent of Internet of Things (IoT) [1,2] and the digital transformation of our society is adding another dimension to the energy issue that fossil fuels and renewables cannot cope with. Powering trillions of sensors [3,4] amounts to a good deal of energy, but each one of those sensors will just require a tiny bit of it to be functionally autonomous. Batteries are currently providing this micro-energy autonomy. However, replacing billions of batteries will not be feasible logistically in near future IoT scenarios and their disposal will generate a huge environmental impact. The renewable alternative to batteries is energy harvesting, by which tiny to moderate amounts of energy can be retrieved from environmental sources such as light, heat or electromagnetic radiation [5].

Thermoelectric generators are a completely solid-state robust approach to energy harvesting featuring no moving parts [6,7,8] that can provide energy autonomy in those IoT settings where a waste heat source is available [9,10,11]. Mainstream thermoelectric technology exhibits some disadvantages in terms of adequacy of materials (toxicity and availability) and assembly complexity (especially when combined with aggressive miniaturization). One avenue to explore in order to deal with these issues is following the silicon path, which is an abundant and technologically enabling material, taking advantage of the fact that nanostructuring [12] can overcome the otherwise poor thermoelectric properties of silicon, as it improves other materials as well [13].

Differently than vertically (π-shaped) or laterally arranged macro thermoelectric generators in which the thermoelectric legs are aligned to the direction of the existing thermal gradient, silicon-based planar micromachined thermoelectric microgenerators feature a transversal architecture. They resort to MEMS technologies to define thermally isolated structures consisting of thin platforms, under which the silicon has been removed [14,15]. If thermal resistances are properly devised, this allows a thermal gradient to be sustained laterally across the structure from a vertically occurring gradient in the environment. Such core standing platforms need to have structural integrity and to be physically/thermally connected to the surrounding bulk silicon by desirably high thermal resistance material links, either ancillary supporting structures or the thermoelectric legs themselves.

## 2. Materials and Methods

Our team has been researching the way to accomplish such transversal architecture by following a so-called, ‘all-silicon’ approximation, i.e., using only scalable silicon-based processing techniques and related silicon-based materials. In our particular unileg approach [16,17,18,19,20], free-standing arrays of p-type silicon nanowires (NWs) arrays have been chosen as the main thermoelectric material, and the thermoelectric circuit is completed with a metal thin film as second thermoelectric material placed on top of ancillary connecting structures.

Silicon NWs were chosen as thermoelectric material because it is expected that nanostructured silicon show better thermoelectric performance than bulk silicon. In particular, thin silicon NWs were shown to feature a competitive thermoelectric figure of merit (ZT > 0.5) as a result of their nano-spurred lower thermal conductivity [21,22].

Promising as these properties may look like for individual low-dimensional Si (nano)objects, the real challenge is to integrate them with appropriate high density into devices able to provide meaningful amounts of power in realistic thermal conditions. Different arrangements of such nanostructured silicon prone to deliver power in the micro-Watt range have been attempted mostly using top-down approaches carving out nanowires from already existing silicon [23,24,25,26,27]. Our approach shares the same goal but differs by following a bottom-up strategy for the integration of the thermoelectric material, so that no advanced nanolithography tool is required. Indeed, Si NWs can be integrated monolithically in the described structure by means of a CVD-VLS post-processing step. Such process yields dense arrays of aligned crystalline NWs bridging laterally the thermally isolated microplatform and the surrounding silicon bulk in a way that those NWs are electrically and structurally connected with the minimum possible thermal and electrical contact resistance [28,29]. For this to happen, the vertical walls to be bridged need to exhibit a [111] orientation. Longer nanowires will have a higher thermal resistance and will enable larger thermal differences. Obtaining them will be more time-consuming, and sometimes, more difficult due to tapering effects. An interesting engineering approach to that problem is to design an arbitrary long bridging distance between the platform and the surrounding silicon rim and to populate it with appropriated equi-spaced silicon trenches, which will be occupied by silicon NWs at the same time in a single well-optimized CVD process for that reasonable length target. Additional thermal performance improvements have been added to the presented architecture by modifying the ancillary supports for the metal legs in order to increase also their thermal resistance [30,31]. Such ancillary supports were long and narrow bulk silicon bridges in earlier generations and wide, short and thin nitride supports in the later generations. A sketch of the described device, where all these features can be appreciated, is shown in Figure 1. A thorough description of the presented thermoelectric microplatform, and the associated materials and processing options can be consulted in previous papers [16,17,18,19,20]. Further relevant Scanning Electron Microscope (SEM) images and details on the design and fabrication sequence of our latest generation devices can be found in [19].

These architectural and design options allow improving the attainable temperature difference within the device. Increasing it further beyond this point implies acting on the thermoelectric material itself substituting the Si NWs arrays by another material with a smaller intrinsic thermal conductivity. For this reason, we have attempted a similar device with SiGe NWs as a silicon-based material alternative. SiGe is hardly unknown in the thermoelectric arena in bulk or nanowire form [8,13]. It was the material of choice in early versions of NASA radioisotope thermogenerators on board of several space missions and it is still powering Voyager probes decades after their launch [32]. Different nanostructuring routes [8,33] and low dimension form factors such as nanowires [13,34] are currently being researched for this material. SiGe NWs can be grown with the same sort of CVD-VLS processing than Si NWs [35] and possess an intrinsically higher thermal resistance already shown in bulk form due to alloying phonon scattering effects [36,37]. In an attempt to go beyond material characterization, this paper focuses in integrating and assessing the performance of such SiGe NWs using similar Si NWs counterparts as a benchmark at working device level, and tracing back the performance outcome to the thermal, electrical and thermoelectric properties of both types of nanowires.

## 3. Results

Si and SiGe NW arrays were obtained following a similar procedure. In the first place, the devices were seeded with gold nanoparticles following a galvanic displacement method. Second, the seeded devices followed a CVD-VLS process with the conditions of temperature, pressure and gas precursors described in Table 1. These conditions were optimized to allow the perpendicular growth of dense and well-oriented arrays of nanowires in both cases. Table 2 shows the average diameter of the NWs obtained and their surface densities. It can be noted that the same seeding conditions for both types of nanowires yielded similar surface densities but much narrower nanowires were obtained for the SiGe case. This is certainly a parameter of importance in the thermal performance of the nanowires. With respect to the length, both types of nanowires were designed to be 10 μm long when bridging the gap between the vertical silicon trenches of the device, the SiGe NWs requiring a longer growth time to reach this length. Transmission Electron Microscope (TEM) observation and electron diffraction patterns (not shown) reveal the crystalline nature for both types of NWs and a hexagonal faceting. Structural characterization pointed to a 20–30% Ge composition for our SiGe NWs. For such a percentage, a thermal conductivity of 15 W/(m·K) is expected in bulk material [36]. It is always interesting to target the lowest Ge content that provides a meaningful thermal conductivity reduction since Ge precursors are expensive.

Once populated with the nanowire arrays, the devices were characterized for the Seebeck coefficient as a function of temperature. They were placed in an oven at different operating temperatures (20–200 °C), and for each of them, a set of different temperature differences (*ΔT*) was forced across the nanowires using a built-in heater in the suspended platform. This heater is not shown in Figure 1, but it can be appreciated in the graphical material of [19]. The open-circuit voltage (*V_oc_*) was determined at each *ΔT*. This *ΔT* was estimated from the TCR (Temperature Coefficient of Resistance) of the heater element, which was previously calibrated as a thermometer. For each oven operating temperature, a *V_oc_* vs. *ΔT* curve was constructed and the Seebeck coefficient at that operating temperature was obtained from the slope of the curve. Electrical magnitudes are measured nowadays with extreme precision, but a certain error could be expected for the Seebeck coefficient as a result of *ΔT* determination from the heater resistance and its TCR value. Nevertheless, due to the fact that the same sort of device was used for both types of NWs and given the fabrication precision associated to IC processing, such an error is anticipated to be small and having an effect on the same direction so comparisons between Si and SiGe are deemed safe and sound. 

For the particular growth conditions of this work, the Seebeck coefficient of Si NWs is higher than the one of SiGe NWs, as shown in Figure 2. The slope of its evolution with temperature is steeper for the Si NWs, and both are steeper than the one of bulk Si, which was similarly determined in devices that feature (top-down) silicon microbeams across the trenches instead of (bottom-up) Si or SiGe NW arrays. As reported in our previous work [19], but not shown in the figure, the evolution of the Seebeck coefficient with temperature of the SiGe NWs is also steeper than the one shown in the literature for SiGe bulk material. The reason of this different behaviour is still under debate and not relevant for the purpose of this work, but some explanations point to phenomena, such as the suppression of the phonon-drag effect for materials structurally enabling phonon scattering mechanisms [38].

We do not have an independent measurement of the doping level of our nanowires, but a first estimation can be gathered by comparing the Seebeck values we have obtained [29] with the data available of Seebeck versus doping in the literature for Si and SiGe in bulk form [37,38,39,40,41]. Following such an approach one may expect doping levels of 4–9 × 10^19^ cm^−3^ for Si NWs and 0.8–1 × 10^20^ cm^−3^ for SiGe NWs. As a reference, the doping level of the bulk Si parts of the device (device layer of our starting SOI wafers) was about 2 × 10^19^ cm^−3^. Unlike Seebeck coefficient, which is not affected by the dimensional characteristics of the nanowires, this device does not lend itself to estimate the electrical conductivity of individual nanowires. Consequently, no assessment of their power factor has been attempted. Uncertainties regarding the fraction of the resistance corresponding to the NWs-leg and dispersion in diameters and densities of NWs within the trench are expected to affect the value of the inferred conductivity up to an order of magnitude, so any calculation and comparison of such magnitude with reference values would not provide any trustworthy insight.

In order to characterize the microgenerators performance as energy harvesters, I-V (and power) measurements were made. Harvesting mode operation (use of an external hotplate) was chosen rather than test mode operation (use of built-in heaters in the suspended platform) to keep such characterization as close to real conditions and operations as possible. Although a much lower effective *ΔT* across the thermoelectric material is anticipated in harvesting mode than in test mode, harvesting mode lends itself to bring out the different thermal conductance of the materials at play.

Measurements were made on chips placed on top of the heated stage of a Linkam THMS 350V using a silver paste to enhance the thermal contact between both surfaces. While the bottom part of the device is at a high temperature, the top of the device is cooled by natural convection. It was experimentally confirmed that in the current setup the temperature at the hot junction area was close to the hotplate temperature to a few percent units. Measurements were conducted in a 2-wire configuration to mimic the performance in a real application scenario. The steps followed for an I-V measurement started by measuring the *V_oc_* at a given hot plate temperature. The current through the device is then increased in small steps, while monitoring the voltage, and the process is stopped once we reach a zero voltage across the microgenerator (equivalent to a short-circuit condition), which corresponds to the maximum current it can provide (*I_sc_*). Finally, V vs. I curves and P (I·V) vs. I curves are plotted to confirm that, due to the linear dependence of V vs. I, the maximum power corresponds to *V_oc_*·*I_sc_*/4. Again, no electrical noise is virtually expected for the measured electrical magnitudes. However, unlike the previous Seebeck coefficient determination, output voltages in harvesting mode are measured with the device exposed to natural convection. Small ambient fluctuations may affect the internal temperature difference, which will translate into a change of electrical parameters. By analyzing the stability of such measurements over a long period of time (3 h), the electrical consequences of thermal fluctuations were determined to be 10 µV and 1 µA in *V_oc_* and *I_sc_* measurements. Those are low values not compromising the subsequent performance comparisons.

The open circuit voltage (Seebeck voltage) and power measured for the devices with Si and SiGe NWs are shown as a function of the external hotplate temperature in Figure 3a,b. The expected linear dependence of Seebeck voltage and the quadratic dependence of power with temperature can be readily appreciated. It is also evident from the figures that SiGe devices exhibit larger voltages and powers for the same hotplate temperatures. The plots shown in those figures, and from now on, correspond to representative devices with a T3 architecture, i.e., with three consecutive trenches filled with nanowires between the isolated platform and the surrounding silicon rim. This means that in both cases the equivalent length of the nanowires was ca 30 μm. Devices with one to up to four trenches (T1, T2, T3, T4) were fabricated. In all situations, the obtained voltage scaled with the number of trenches, that is, for a given hotplate temperature, a larger *ΔT* was obtained across the nanowires, the longer these were. However, this was not the same for the power, and T4 devices usually showed a lower power than T3 ones. This is because the device electrical resistance also increases as the number of trenches increases, and voltage and resistance have conflicting roles in power. Moreover, the positive effect of nanowire length in *ΔT* tends to saturate since as the nanowires thermal resistance keeps increasing other parallel thermal resistances kick in (metal legs, ancillary platform supports, and air itself) hampering a further increase of *ΔT*.

## 4. Discussion

Since Seebeck voltage (depending on the Seebeck coefficient and the attained internal *ΔT*) and the resistance have an impact in the obtained power, it is worth ascertaining their different contributions in the values obtained for both types of nanowires.

Table 3 sums-up those parameters obtained at a hotplate temperature of 200 °C. Seebeck voltage and device resistance are the primary measurements. Just to compare Si and SiGe NWs in terms of effective power factor, a ‘device power factor’ can be assayed using the device resistance instead of electrical conductivity. At 200 °C, this magnitude is 7.7 nW/K^2^ for Si NWs and 2.7 nW/K^2^ (3.5 times lower) for SiGe NWs.

Power itself is derived from the device I-V curves as V·I at matching load conditions (i.e., at its maximum), and power density is calculated assuming an effective device area of 2 mm^2^. This area value attempts to reflect the effective footprint of the device as sketched in Figure 1, keeping in mind that such device will be laterally replicated in series or parallel configurations in order to build a full microgenerator. For this reason, this compromise value is larger than the area occupied by the platform/trenches/membrane (roughly 1.3 mm^2^) and smaller than the one including the full contacts (roughly 2.6 mm^2^), since these elements will be partly shared with the neighboring devices.

The temperature difference between the hot and cold end of the device (*∆T*) can be estimated from the open circuit voltage and the previously measured Seebeck coefficients and it is represented in Figure 4. It is worth noticing that in harvesting conditions the internal *ΔT* is rather low so the overall temperature of the nanowires is not much different from the hot junction one. Thus, it is safe to use the Seebeck coefficient at the corresponding hotplate temperature for the first estimation of internal *ΔT* in our devices.

The overall thermoelectric performance of a given device depends on the thermoelectric, electrical and thermal properties of its material arrangement, that is, on the Seebeck coefficient, electrical resistance and thermal conductance. It is clear from Figure 3 that the improvement brought by SiGe NWs is outstanding despite this material not excelling, neither in the Seebeck coefficient nor in electrical resistance when compared to Si NWs.

Certainly, when replotting the power results as a function of the open circuit voltage, Si NWs values are above the ones of SiGe, signaling a lower device resistance, as shown in Figure 5a. When replotting power as a function of internal *ΔT*, as in Figure 5b, the outperformance of silicon is even more extreme because its higher Seebeck coefficient, and so open circuit voltage, adds to its lower resistance value. Of course, this only holds for the limited range of *V_oc_* and *ΔT* that both Si and SiGe NWs share due to thermal underperformance of Si. The advantage of SiGe is that its better intrinsic thermal properties extend the operating window of *V_oc_* and *ΔT* beyond what is possible for Si NWs for the same hotplate temperatures range.

Unsurprisingly, the big advantage of SiGe comes from its lower thermal conductivity, which in bulk form is already one tenth of that of silicon. Nanostructuring may have probably decreased further this value, as it is the case for silicon: thermal conductivities of Si NWs similar to the ones of this study were measured to be 15–30 W/(m·K) [29], e.g., 4 or 5 times lower than the bulk value. Further work is ongoing to measure the thermal conductivity of the SiGe NWs, but the starting point of bulk SiGe thermal conductivity is already similar or even lower than the one of Si NWs, so much lower values are to be expected. Moreover, the nanostructuring effect for our SiGe NWs may be more severe than for our Si NWs given the much smaller diameter of the former, so a more aggressive reduction of their thermal conductivity may be possible and conductivities in the order of 1 W/(m·K) would be expected [42]. In any case, thanks to this situation, the devices with SiGe NWs are able to capture a much larger portion of the available thermal gradient as shown in Table 3 and Figure 4. Up to 15 °C, ten more times than for Si NWs, are available to be converted into electricity at a hotplate temperature of 200 °C.

Improving further the performance of SiGe NWs devices may be challenging by the interdependency of material properties at play. Increasing SiGe Seebeck coefficient would require decreasing the doping, which will lead to a larger electrical resistance (already high according to Table 3 values), thus, inhibiting the increase in power. Decreasing the resistance would demand increasing the doping, but that also would decrease of the Seebeck coefficient leading again to a stalemate in terms of power. Alternatively, increasing the diameter of the nanowires or their density will decrease the electrical resistance. Although this option will also decrease the thermal resistance potentially decreasing the internal *ΔT* and the obtained open circuit voltage, it holds some latitude for power improvement since the thermal resistance of SiGe NWs is so high that the overall device thermal resistance could be starting to be limited by the one of other (ancillary) components as explained earlier. Therefore, a fraction of SiGe good thermal conductivity could be traded for a lower electrical resistance, and thus power, without affecting much negatively the overall thermal performance of the device.

There is one remaining issue worth noticing. In general, the effective *ΔT* attained across the nanowires is a rather small fraction (less than 10% in the very best case) of the *ΔT* that is present in the environment. This is because it is very hard to dissipate effectively heat by conduction/convection in the cold side of the device, due to its small dimensions in the millimeter regime. Increasing the attained *ΔT* requires decreasing the thermal resistance to the ambient by means of forced convection or the integration of a heat exchanger [43,44], which is not always possible in real applications in the first case, and multiplies the volume of the device in the second case. It is worth noting that for the SiGe NWs devices, a power density of 7 μW/cm^2^ has been attained in real harvesting conditions without any heat exchanger, so that a usable power density is at reach from a really minimal size device in terms of volume.

## 5. Conclusions

Silicon and silicon-germanium arrays of nanowires have been physically and functionally integrated in a silicon micromachined structure in order to obtain a monolithic and compact silicon-based thermoelectric microgenerator. The device operates as a unileg thermocouple with an extended nanostructured semiconductor leg completed with a thin film metal leg, and it successfully exploits the enhanced thermoelectric properties of nanostructured silicon materials. MEMS (Micro-ElectroEechanical Systems) processes are used to produce an architecture that confers to the device the adequate thermal design to convert a vertical thermal gradient into a lateral one. The device lends itself to affordable large replication schemes thanks to its compatibility with planar silicon technologies and the smart combination of top-down and bottom-up technological processes that precludes the use time-consuming nanolithography. Open circuit voltages in the mV range and power densities in the μW/cm^2^ range, compatible with future IoT demands, have been obtained for such a single thermocouple device architecture under real harvesting conditions (no artificially forced thermal gradient). Although Si nanowires show better electrical (resistance) and thermoelectric (Seebeck) properties, devices with SiGe nanowires show drastically better voltage and power results in the temperature range explored (50–200 °C). This is possible thanks to the much better thermal properties (lower thermal conductivity) of SiGe material that allow developing a much larger internal temperature difference across the device when harvesting heat under natural convection conditions and no heat exchanger. Despite this achieved improvement, further optimization is yet to come. The fine-tuning of doping, diameter and germanium composition can potentially improve further the performance of the presented material trading part of its optimum thermal performance for a better electrical one.

In conclusion, the use of MEMS compatible nanowires of silicon-germanium instead of silicon nanowires showed an overall improvement in the performance of micro thermal harvesters and reached usable power densities in a minimal volume device (without heat exchanger), adding a remarkable potential to this silicon-based fabrication route in the onset of a massive deployment of self-powered IoT devices.

## Figures and Tables

**Figure 1 nanomaterials-11-00517-f001:**
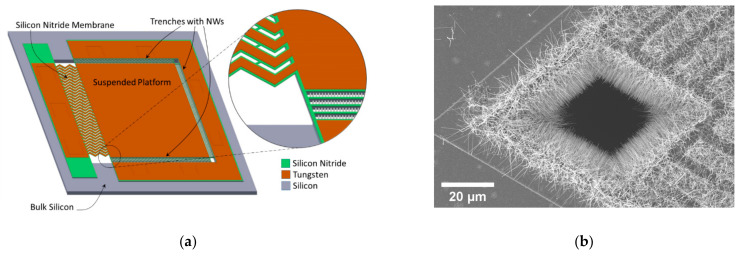
(**a**) Sketch of the fabricated devices consisting of a suspended (typically 1 mm^2^) platform connected to the bulk part of the device by a thin structured nitride membrane on top of which the metal leg is placed, and by Si-based nanowires (NWs) along the other three sides of the platform. In these regions, the NWs bridge a given number of 10 μm wide trenches (from 1 to 4 in the current design) defined by micromachined vertical Si walls. These Si-based NWs arrays configure the laterally extended nanostructured semiconductor leg of the thermocouple; (**b**) SEM image of the device corner opposite the membrane where grown dense arrays of nanowires can be appreciated bridging the trenches and growing freely inside the corner cavity.

**Figure 2 nanomaterials-11-00517-f002:**
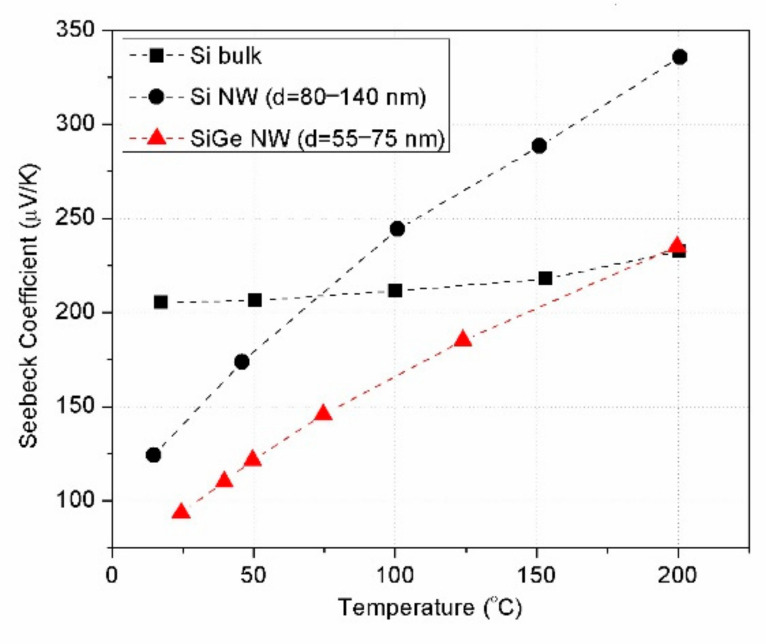
Seebeck coefficient as a function of temperature measured in devices with Si NWs legs, SiGe NWs legs, bulk Si legs bridging the suspended platform and its surrounding Si rim.

**Figure 3 nanomaterials-11-00517-f003:**
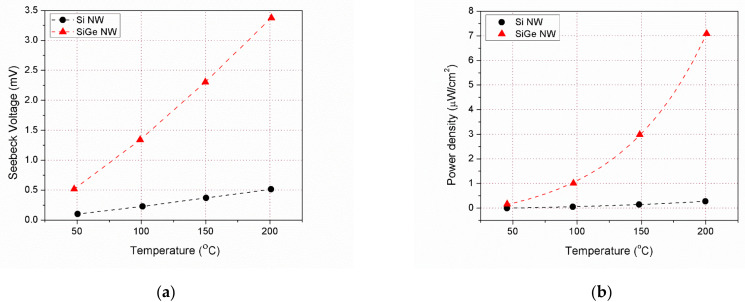
(**a**) Seebeck voltage (open circuit voltage) as a function of hotplate temperature measured for Si and SiGe NWs devices; (**b**) Maximum electrical power density obtained in both types of devices in the same temperature conditions.

**Figure 4 nanomaterials-11-00517-f004:**
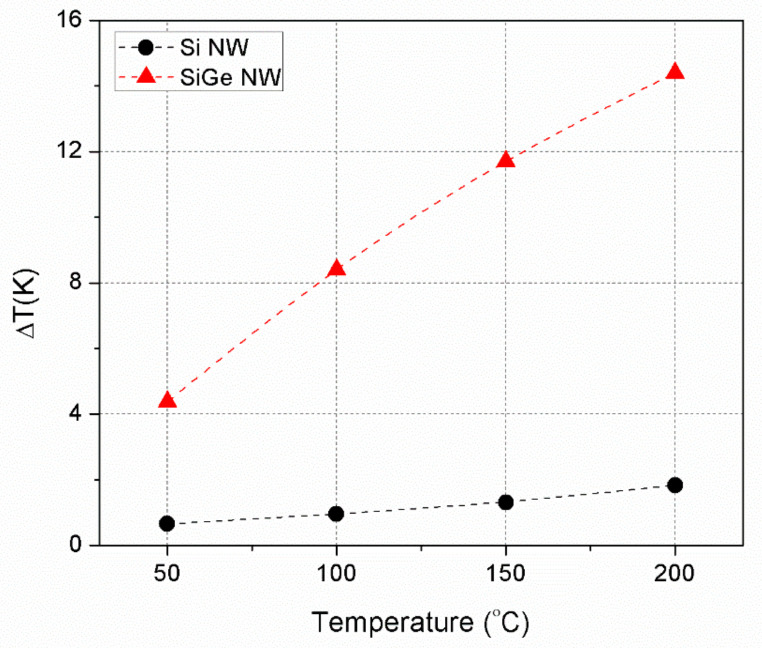
Estimated *ΔT* across Si and SiGe NWs inferred from the Seebeck coefficients and the open circuit voltages measured for them as a function of the hotplate temperature.

**Figure 5 nanomaterials-11-00517-f005:**
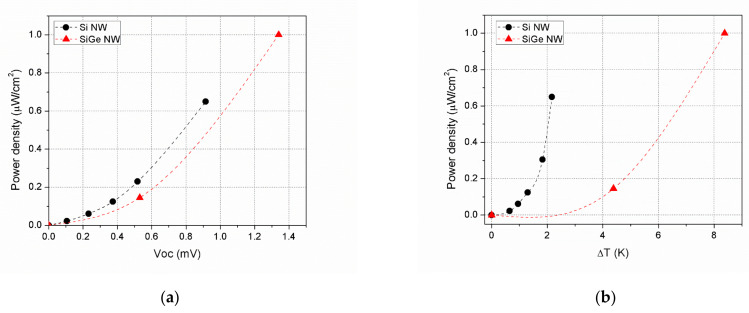
(**a**) Maximum power density obtained for Si and SiGe NW devices as a function of the open circuit voltage (showing the different device resistance at play); (**b**) internal *ΔT* (showing the different electrical resistance and Seebeck coefficient at play). Notice that for clarity, only the data corresponding to the lower *ΔT* portion available for SiGe has been represented.

**Table 1 nanomaterials-11-00517-t001:** VLS-CVD growth parameters for Si and SiGe NWs.

Parameter	Si NWs	SiGe NWs
Temperature	630 °C	650 °C
Pressure	2.5 Torr	2.5 Torr
HCl	30 sccm	30 sccm
H2	1000 sccm	1000 sccm
SiH4—H2 (10%)	150 sccm	200 sccm
GeH4—H2 (10%)	-	8 sccm
B2H6—H2 (750 ppm)	50 sccm	50 sccm

**Table 2 nanomaterials-11-00517-t002:** Resulting properties of Si and SiGe NWs after their VLS-CVD growth.

Property	Si NWs	SiGe NWs
NW diameter	112 ± 31 nm	64 ± 11 nm
NW density	3.9 NWs/µm^2^	4.9 NWs/µm^2^

**Table 3 nanomaterials-11-00517-t003:** Harvesting parameters for three trenches devices populated with Si or SiGe NWs for a hotplate temperature of 200 °C and an effective device area of 2 mm^2^.

Property	Si NWs	SiGe NWs
R_device_ (Ω)	15.06	19.50
*V_oc_* (mV)	0.52	3.37
∆*T* (°C)	1.54	14.4
Max. Power Density (μW/cm^2^)	0.23	7.10

## Data Availability

The data presented in this study are openly available in Digital.CSIC at http://dx.doi.org/10.20350/digitalCSIC/13743.

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
