# Peer review of "Transitioning from Si to SiGe Nanowires as Thermoelectric Material in Silicon-Based Microgenerators"

_nanomaterials, 2021, doi:10.3390/nano11020517_

Round 1

Reviewer 1 Report

The authors introduce the transitioning from Si to SiGe nanowires as thermoelectric material in silicon-based microgenerators. After carefully going through this manuscript, I found this work is interesting to some extent, and the evidences can well support their results. The structure of the work is well organized with a clear logic, and the figures have good quality. However, there are still some unclear points that need to be answered or solved, therefore at least a major revision is needed before publishing in this journal. My comments may help the authors further improve their work:

  1. In the introduction part, the progress and current challenge of Si and SiGe nanowires are not well summarized. Please enrich this part with a clear logic. Some comprehensive review works related to this research topic should be helpful and cited in appropriate cites, such as Chem. Rev. 2019, 119, 15, 9260–9302 and Chem. Rev. 2020, 120, 7399−7515 (chapter 4.2.2).
  2. For the characterization of the achieved nanowire, the authors only provide their SEM image as shown in fig 2b (the SEM information below the image should be removed). This is not enough. Since the nanowires should be crystalline materials, at least the authors should provide the TEM and HRTEM image to confirm such crystalline feature.
  3. All measured results need error bars, and the authors should state how they determine the error bars.
  4. Why there is only Seebeck coefficient results but no electrical conductivity and power factor?
  5. The conclusion part needs revision to show more clear highlights.

Author Response

Please, see attachment

Reviewer 2 Report

The Authors reported a silicon-based microgenerators device based on SiGe nanowires, it is a very interesting TE micro device and high performance. But this still has some points that need the author to indicate and explain the following.

  1. From figure 1 diagram, I understand the SiGe nanowires growth between the suspended platform and bulk silicon. So ΔT is also between the bulk silicon on a hotplate and the suspended platform, which no contact with the hotplate, is it correct?
  2. Table 2, on 200 degrees hotplate, why the ΔT of Si NWs only 1.54, too low in here, and SiGe NWs is 14.4. Why the SiGe close 10 times than Si NWs? Did the authors measure the thermal conductivity of Si and SiGe Ns?
  3. The author needs to give clearly the description of the 2 mm2 device area since the microdevice in parallel with the hotplate. Is this device area suspended platform area, trenches area, or bulk silicon area?
  4. In Figure 2, the authors need to give the electrical conductivity data with temperature, which is important to characterize the power factor or the SiGe NWs.
  5. This is very interesting design and work, but the low power output since the device area too limit and the temperature different hard to increase. I think it is the problem of the heat flow parallel to the hotplate. If the microgenerator can design like sandwich, the NWs is the growth between the suspended platform and silicon substrate, the performance will increase 10-100 times now.

Author Response

Please, see attachment

Reviewer 3 Report

This work is interesting for the thermoelectric community. It is thoroughly performed and presented. The conclusion is scientifically and technologically valuable. 

The figure caption of Figure 1 should be corrected.

It would be nice if the authors could comment on the following:

SiGe shows better performance in the actual architecture due to the lower thermal conductivity of SiGe compared of that of Si. NWs properties are not optimized against doping and thermal conductivity. Would SiGe show better performance than Si in devices based on optimized Si/SiGe NWs ?

Author Response

Please, see attachment
